# Food Insecurity Prevalence, Severity and Determinants in Australian Households during the COVID-19 Pandemic from the Perspective of Women

**DOI:** 10.3390/nu13124262

**Published:** 2021-11-26

**Authors:** Sue Kleve, Christie J. Bennett, Zoe E. Davidson, Nicole J. Kellow, Tracy A. McCaffrey, Sharleen O’Reilly, Joanne Enticott, Lisa J. Moran, Cheryce L. Harrison, Helena Teede, Siew Lim

**Affiliations:** 1Department of Nutrition, Dietetics and Food, School of Clinical Sciences, Monash University, Melbourne, VIC 3168, Australia; christie.bennett@monash.edu (C.J.B.); zoe.davidson@monash.edu (Z.E.D.); nicole.kellow@monash.edu (N.J.K.); Tracy.McCaffrey@monash.edu (T.A.M.); 2School of Agriculture and Food Science, University College Dublin, Belfield, D04 V1W8 Dublin, Ireland; sharleen.oreilly@ucd.ie; 3Monash Centre for Health Research and Implementation (MCHRI), School of Public Health and Preventive Medicine, Monash University, Melbourne, VIC 3168, Australia; Joanne.Enticott@monash.edu (J.E.); lisa.moran@monash.edu (L.J.M.); Cheryce.Harrison@monash.edu (C.L.H.); helena.teede@monash.edu (H.T.); siew.lim1@monash.edu (S.L.)

**Keywords:** food security, women, mental health, Australia

## Abstract

This study aimed to describe the prevalence, severity and socio-demographic predictors of food insecurity in Australian households during the COVID-19 pandemic in 2020, from the perspective of women. A cross-sectional online survey of Australian (18–50 years) women was conducted. The survey collected demographic information and utilised the 18-item US Department of Agriculture Household Food Security Survey Module and the Kessler Psychological Distress Scale (K10). A multivariable regression was used to identify predictors of food security status. In this cohort (*n* = 1005), 19.6% were living in households experiencing food insecurity; with 11.8% experiencing low food-security and 7.8% very low food-security. A further 13.7% of households reported marginal food-security. Poor mental health status (K10 score ≥ 20) predicted household food insecurity at all levels. The presence of more than three children in the household was associated with low food-security (OR 6.24, 95% CI: 2.59–15.03). Those who were renting were 2.10 (95% CI: 1.09–4.05) times likely to experience very low food-security than those owning their own home. The COVID-19 pandemic may have contributed to an increased prevalence of household food insecurity. This study supports the need for a range of responses that address mental health, financial, employment and housing support to food security in Australia.

## 1. Introduction

Food security is a fundamental human right and paramount to physical, mental and social health and wellbeing [1]. Yet food insecurity, defined as ‘the limited or uncertain availability of nutritionally adequate and safe foods, or the ability to acquire acceptable food in socially acceptable ways’ is increasing in high income countries such as Australia [2]. Food insecurity at a household and individual level is characterised by episodic and/or chronic experiences of stress, anxiety, concern, social isolation and compromise to the quantity and nutritional quality of food [3,4]. In 2011, using a two-item measure, 4% of Australians lived in a household that reported to be food insecure [5]. Key determinants of food security status include income level, income shocks, available economic resources for purchasing food and general resources in a household [6,7]. Some population groups in Australia may be at greater risk of experiencing food insecurity. For example, individuals experiencing material and/or financial hardship, Aboriginal and Torres Strait Island peoples, people from a Culturally and Linguistic Diverse background including refugees and people seeking asylum, single-parent households and people experiencing homelessness [6,8,9,10,11,12]. Due to the financial consequences of the SARS-COV2 (COVID-19) pandemic in Australia, the prevalence and severity of food insecurity may be greater than previously documented, especially within at-risk groups.

COVID-19 was first identified in Australia in January 2020 [13]. Australia consists of eight states and territories, each with individual governments who were responsible for the management of the pandemic response in each respective jurisdiction. COVID-19 related restrictions (lockdowns including business and school closures, travel restrictions, wearing masks and social distancing requirements) therefore varied across Australia throughout 2020 due to varying case numbers in states and territories (Figure 1). Although COVID-19 case numbers were comparatively low in a global context, some states such as Victoria endured significant periods of lockdown [14]. The federal government provided a variety of financial support payments to buffer household income (Jobkeeper and Jobseeker). Despite the multiple economic stimuli initiated in 2020 [14], the direct and indirect impacts of COVID-19 such as lockdowns, working from home requirements, schooling from home, loss of regular social support structures, child care closure and loss of employment and income placed many households under additional stress [15]. Women were more likely to shoulder the extra domestic burden and experienced this additional pressure on top of existing household and food provisioning demands [16].

Understanding the food experience of women is particularly important due to their unique role in the household and society. Despite increasing participation by women in the workforce, either in casual, part-time or full-time work, many continue to have the primary role and investment of time in household and family food provisioning [16]. As a result of the understanding and experiences of food provisioning roles, it is suggested that women in food insecure married/partnered households may report higher levels and experiences of food insecurity [18,19]. Unsurprisingly, due to the increased load, women have exhibited higher levels of stress during the pandemic [20]. Therefore, understanding the food insecurity experience from the perspective of the female in the household during the COVID-19 pandemic is important as women are often the gate keeper to food security.

Recognition of the links between food insecurity, poor diet quality, poor mental health and chronic disease development have prompted calls for more research exploring the impact of the COVID-19 pandemic on food security worldwide [21]. The few published studies characterising the effect of COVID-19 on food insecurity have found conflicting results, with both improvements [22] and reductions [23,24] in access to food reported. While these studies explored the impact of the pandemic on food security changes in the general population, there is a need to focus research specifically on groups at potential greater risk of food insecurity within society, such as women and children. Therefore, the aim of the study was to determine the prevalence, severity and socio-demographic predictors of household food insecurity from the perspective of women of reproductive age (18–50 years) in Australia during the COVID-19 pandemic.

## 2. Materials and Methods

### 2.1. Study Design

This study was part of a larger study to explore food intake, physical activity and mental wellbeing during COVID-19 pandemic in women of reproductive age (18–50 years) in Australia. This national cross-sectional online survey was undertaken between 15 October to 7 November 2020 during COVID-19 (see further detail below). This research captured a large sample of the female Australian population across age and residential location (state/territory and remoteness area).

### 2.2. Participants and Sampling Strategy

Women of reproductive age (age 18–50) who resided in Australia were recruited by an external cross-panel market research provider (Online Research Unit) with a well-established database of 400,000 members. This research provider uses multiple recruitment methods (telephone, online, print and postal) [25]. Men or women under the age of 18 or over 50 were ineligible to participate. Participants were invited to complete the online survey via targeted emails describing the content and duration of the survey. The proportion of women of reproductive age from each state and territory were recruited according to the Australian Bureau of Statistics (ABS) population characteristics [26]. Whilst this cannot be representative across all population characteristics, it is a widely accepted approach and the recruitment was designed to obtain a sample consistent with the population proportions across age, gender and residential location (state/territory) [27]. On day four and five of the survey distribution, location of residence (state/territory) of respondents were examined, and further sampling was targeted to underrepresented groups to align with population characteristics. Respondents were reimbursed in line with ISO 26362 industry requirements, and reimbursements were mailed to a residential address inside Australia, therefore ensuring that respondents were living in Australia. All data collected was anonymous. Participants provided online consent after reading the study purpose, before participating in the 10-min online survey. The study was approved by the Monash University Human Research Ethics Committee (HREC project: 25941).

### 2.3. Survey Variables

The survey included multiple-choice questions to assess respondents’ age group, cultural or ethnic group, highest level of education completed, employment status before the pandemic, changes in employment status during the pandemic and annual household income before tax pre-pandemic and during pandemic at the time of sampling in 2020. Urban or rural/remote location was determined based on postcode.

Food security status was assessed using the validated 18-item United States Department of Agriculture Household Food Security Survey Module (USDA-HFSSM), including 10 adult questions and 8 child questions [28]. The USDA-HFSSM was selected for determination of food security status because of its reliability across populations and population subgroups; and its ability to capture the severity level and continuum of experience of food security. Survey respondents were asked to consider the previous six months when answering questions to capture experience during the COVID-19 pandemic. The USDA-HFSSM protocol for households with children uses the number of affirmative responses to the 18 questions to provide a raw score categorising households’ food security severity: high food security (score of 0) with no reported indications of food-access limitations; marginal food security (score of 1–2) indicating anxiety over food sufficiency or a shortage of food in the house; low food security (score of 3–7) indicating reduced quality and variety of food with little or no indication of reduced intake; and very low food security (score 8–18) describing multiple indications of a disrupted eating pattern and reduced food intake. For data analyses HFSSM protocol identified four categories of food security: high food security, marginal food security, low food security and very low food security.

The Kessler Psychological Distress Scale (K10) was used to assess psychological distress [29]. The K10 is a validated 10-item questionnaire to measure level of distress based on questions about anxiety and depressive symptoms experienced by an individual over the last 4 weeks. A score <20 was categorised as likely to be mentally well, and ≥20 was considered likely to have mental health concerns [29,30].

### 2.4. Statistical Analysis

Data screening and cleaning ensured data usability and an integrity script allowed discarding of surveys with less than 10% completion (*n* = 508). These non-completions are mostly due to ineligibility (e.g., being male) as determined in the screening questions at the start of the survey. Descriptive statistics were calculated for all variables. Chi-squared was used to compare demographic, socioeconomic, mental health and food security variables between the four food security severity groups. Multivariable logistic regression was conducted. Covariates used in the multivariable model were those identified using descriptive statistics to be significant between food security severity groups. Covariates included state, marital status, children, education, pre-COVID-19 employment, pre COVID-19 income and change to employment status due to COVID-19. Descriptive statistics were used to report the responses to the USDA-HFSSM 18-items across the four food security groups. Data were analysed using the statistical software package IBM SPSS for Windows Version 26 (SPSS INC., Chicago, IL, USA). Significance was set at *p* < 0.05.

## 3. Results

### 3.1. Demographic Characteristics

A total of *n* = 1005 women were included in the analysis. Results are presented in accordance with the four categories of food security severity status. Two thirds of respondents were categorised as living in a household experiencing high food security (*n* = 670, 66.7%), 13.7% (*n* = 138) marginal food security, 11.8% (*n* = 119) low food security and 7.8% (*n*= 78) experiencing very low food security.

Survey respondent socio-demographic characteristics across the four food security categories are presented in Table 1. The majority of respondents (*n* = 847, 84.3%) reported living in a major city of Australia, aged 25–44 years (*n* = 678, 67.5%) and in a married or de facto relationship (*n* = 574, 57.1%). Forty-three percent of respondents (*n* = 431) reported having one or more children. With regard to education status, over half (*n* = 561, 55.8%) had a university education (Bachelor’s degree or higher) and 42.7% (*n* = 429) reported a household income greater than AUD 100,000 per year [31]. Households who were classified as having low and very low food security were more likely to report mental health concerns (*p* < 0.001), were more likely to be renting than owning their own home (*p* = 0.006), be single (*p* < 0.005), have children (*p* < 0.001) and were less likely to have post-secondary education (*p* = 0.003) compared to households that were food secure.

### 3.2. Food Security Status

The proportions of responses to the 18-item USDA-HFSSM questions according to each food security category are presented in Table 2. This data highlights the experience of food security across the different severity categories. Women who were in marginally food secure households reported that they worried whether food would run out before they had money to buy more. As the questions increased in the severity of experience, there was an increase in frequency of respondents who were classified as low and/or very low food secure. For example, in response to the question “*did you or other adults in your household ever cut the size of your meals or skip meals because there wasn’t enough money for food*”, those that responded affirmatively were experiencing low (31.5%) or very low food security (67.6%). Of households experiencing very low food security, 79.7% reduced the size of their meals either every month or 3–4 months. Similarly, 95% of respondents experiencing very low food security reported in the last 6 months that either they or other adults in the household had not eaten for a whole day because there was not enough money for food. Households responding to the initial child experience question reported that they relied on only a few types of low-cost food items to feed children because they were running out of money to buy food (28% marginal food secure, 44.8% low food security and 27% very low food security). However, as the questions increased in the severity of food security experience those with low and very low food security status were more likely to respond affirmatively.

### 3.3. Factors Associated with Food Security

Socio-demographic characteristics associated with food security status are reported in Table 1. As food insecurity severity increased, women were more likely to be employed casually, be on government disability assistance payments or a homemaker (*p* = 0.014) prior to the COVID-19 pandemic. There was an association between food security and changes in employment status due to COVID-19, whereby those that reported changes in their employment due to COVID-19 were more likely to experience food insecurity (*p* < 0.001). There was also an association between severity of food security and level of income. Women in the lowest household income brackets (<AUD 50,000) were more likely to experience very low food security (*p* < 0.001) pre-COVID-19. While not statistically significant, there was a trend for Victoria and New South Wales, the States with the largest population size and with the highest number of COVID-19 cases, to report the highest proportion of low and very low food security (*p* = 0.058).

Table 3 presents the multivariable analysis exploring predictors of food security. As the household pre-COVID-19 income increased, the risk of food insecurity decreased but was still evident. Respondents with a pre-COVID household income between AUD 0–24,999 were more likely to experience marginal (crude odds ratio (OR) 8.90, 95% confidence interval (CI): 3.13–25.30), low (OR 10.29, 95% CI: 2.84–37.36) or very low food security (OR 8.98, 95% CI: 2.18–37.03). However, those with a pre-COVID-19 household income of AUD 100,000–124,000 still had an increased risk of experiencing low food security (OR 3.54, 95% CI: 1.36–9.24). Change of employment status as a result of COVID-19 during 2020 increased the likelihood of experiencing very low food security (OR 6.51, 95% CI: 3.48–12.21).

Poor mental health status (K10 score ≥ 20) of respondents was a predictor of food insecurity at all severity levels. Furthermore, women living in a household experiencing very low food security had seven times higher odds (OR 7.07:95% CI 3.40 -14.66) of having poorer mental health. Other characteristics associated with food insecurity included children and housing status. The presence of more than three children in the household was associated with low food security (OR 6.24, 95% CI: 2.59–15.03). Those who were renting were 2.10 (95% CI: 1.09–4.05) times likely to experience very low food security than those owning their own home.

## 4. Discussion

This study describes the severity of household food security experienced by Australian women during the COVID-19 pandemic in 2020. Our results indicate that, between May and early November 2020, one in five (20%) women were experiencing food insecurity. Of these, 11.8% were experiencing low food security and 7.8% were experiencing very low food security, the most severe form of food insecurity. In addition, 13.7% of respondents were living in households experiencing marginal food security, defined as the early stage of the food security continuum, including experience of stress and anxiety about running out of food. The prevalence of food insecurity was considerably higher (i.e., 5-fold increase) than the reported national prevalence of households (4% [5]) and of women (3.9% [12]) aged over 19 years in 2011–2012, who reported that in the last 12 months they had run out of food and could not afford to buy more [12]. According to the corresponding question in the USDA-HFSSM used in our survey, 14.8% of women responded affirmatively, indicating that food insecurity was significantly higher than reported prior to the COVID-19 pandemic.

Food insecurity prevalence and related socio-demographic factors associated with COVID-19 has been explored across high income countries. The majority of these studies have explored food insecurity specifically in the adult population. Another Australian cross-sectional study conducted in the state of Tasmania (*n* = 1170 adults) between May-June 2020 found that 26% of the sample were experiencing marginal, low or very low food security, greater than that previously reported pre-COVID-19, particularly among economically vulnerable households and people who had lost income during the pandemic [24]. This study used the USDA-HFSSM six-item short form of the 10 adult items in the USDA-HFSSM 18-item [24]. A cross-sectional study, including a national panel of US residents (*n* = 10,368), with post-stratification weighted by gender, age, race, income, and geography (state), were surveyed using the 10-item USDA adult food security module in March 2020 to explore COVID-19 related impacts on food security status. More than one-third of respondents reported to be food insecure. The increase in food insecurity was reported across the general population and higher risk groups comprised of low income and minority populations [23]. Results from a study of the US state of Vermont, reported a 33% increase in the experience of food insecurity since the beginning of the pandemic, to 24% of surveyed households, who indicated disrupted eating [32]. A survey undertaken by Statistics Canada during April-May 2020 found that one in seven (14.6%) Canadians experienced food insecurity in the previous 30 days, with 2% reporting the most severe form of food insecurity [33]. Those who experienced COVID-19 related reductions in employment were more likely to be food insecure (28.7%) than those who were working (10.7%). Consistent with our findings, a higher rate of food insecurity was reported among Canadians living in a household with children (19.2%) compared to those living with no children (12.2%). Since the commencement of lockdown, a UK study reported a twofold increase, where 16.2% of adults had experienced food insecurity. [34]. Consistent with our findings on the severity of experience it was reported that adults skipped meals, and went without food as a result of food insecurity [34]. These studies support the findings of the impacts of COVID-19 on household food security in both adults and children.

This is the first Australian study to explore the experience of food insecurity during the COVID-19 pandemic in households with children using the validated 18-item USDA-HFSSM that includes eight child focused questions. As described in previous studies, the food security status of adults in the households with children may be compromised in preference to that of the children, where possible adults employ child focused protective strategies. This may be in the form of a managed process attempting to shield children from the experience but also to conceal the parent’s own experience [3,6,35,36]. Twenty five percent (*n* = 108) of women with children in the current study were living in households experiencing food insecurity, of these 68% were low food security and 32% very low food security. An additional 17% (*n* = 72) of households with children were classified as experiencing marginal food insecurity. Concerningly the responses to the child experience questions of the USDA-HFSSM (Table 3) indicated that some participants’ households experience of food insecurity was impacting their children, with changes to type and amount of food available for children. National monitoring of the prevalence of food insecurity among children in Australia does not occur, but available data indicate that the number of children living in households at risk of or experiencing food insecurity may be high even before the pandemic [37,38]. A pre-COVID Western Australian survey found that 80% (*n* = 100) of socioeconomically disadvantaged families were experiencing low or very low food security [39]. This survey also highlighted the coping strategies of these families, 67% of adults reported they were unable to feed their children a balanced meal due to limited finances, 27% had to reduce the size of their children’s meals and 13% reported that at least one of their children needed to skip a meal because there was not enough money for food [39]. In 2021, Valardo et al. qualitatively explored the experience of food insecurity amongst 11 South Australian children aged between 10–13 years old, whose households were experiencing socioeconomic disadvantage pre-COVID-19 [40]. Children reported the implications of restrictions in the amount of food available at home and how the quantity and quality of food available could vary depending on household finances [40]. Children who grow up in food insecure households have higher rates of acute (including emergency department admission) and chronic health conditions (such as asthma and allergies) [41] and the experience can impact social, emotional and mental development [42]. While both of these studies are in lower socioeconomic households, our evidence suggests that food insecurity is also present in households beyond those of very low incomes. The mental strain of this compromise and guilt has been reported to be a significant strain for parents in food insecure households [6]. This again highlights the bi-directional relationship between mental health and food security. Additional stress related to the COVID-19 pandemic, including job and financial uncertainty and increased responsibilities such as home-schooling, may have further exacerbated this relationship [43]. Therefore, when exploring food insecurity at a household level, it is important to understand both the adult and child experience.

The results of this study show that mental health, housing tenure, income level and change of employment status due to COVID-19 impacted food security. It has been well documented that financial stability, housing tenure and mental health and food security are interrelated factors. A recent US study showed that during the COVID-19 pandemic, having assets and a lack of financial stressors were important factors in maintaining good mental health [44]. For example, owning assets, such as a house, reduced the risk of mental health issues [44]. A bidirectional relationship between mental health and food security has been well described prior to the pandemic [45]. Further, while there was a relationship between income level and food security, food insecurity was experienced across the income spectrum as previously reported [6,24]. Even for those within the median income of Australian households (AUD 75,000–9999) there was an increased risk of experiencing food insecurity. It is important to consider that income as a single static variable is not an appropriate predictor of food security status as this variable does not account for the regularity of income, budgeting skills or expenses. Further, assets ownership (including home and vehicles) is not good predictor of household financial status as it does not explore the asset to debt ratio, which has significant implications on food security [46]. Across income levels, savings may be used in times of economic stress to protect against food security, which was not captured by this study [46,47]. Despite increasing participation by women in the workforce either in casual, part-time or full-time work, many continue to have the primary role and investment of time in household and family food provisioning [16]. It was clear that the change in employment status was also a significant factor in food security. However, due to statistical power, this could not be further interrogated within this cohort.

Interestingly, despite the varied case numbers, lockdown and changes in employment as a result of pandemic restrictions, there were no differences in the prevalence of food insecurity between Australian states and territories. This could be because financial support provided by the Federal Government was available nationally. The Federal Government implemented a range of economic responses in an attempt to support and protect individuals’ and households’ income [14]. These initial economic relief measures included JobKeeper, a payment made to businesses to maintain employee employment and the Coronavirus Supplement, an increase in government support payments, including for those unemployed called Jobseeker. This addresses the primary determinant of food security. Therefore, the financial impacts of the extended lockdowns in the state of Victoria may have been buffered by the Federal support [14]. This may change as Australia continues to experience strict restrictions, which have impacted industries such as retail, beauty, recreation and hospitality without financial support from the Federal or State Governments [48]. Another reason we may not have seen differences between the states may be due to the sample size of this study, which was not powered for state-based comparisons.

A strength of this study is that it is one of the few studies to explore the impact of the COVID-19 pandemic on food insecurity in Australian households, with a specific focus on the experience of women and children. Furthermore, this national sample using the 18-item USDA-HFSSM has provided a more comprehensive insight beyond the current population one or two-item food security measure to the severity of the food insecurity experience, including perturbations to food intake and coping strategies. This tool also asks about the experience of other adults in the household. Importantly, the use of a more comprehensive tool containing the 8-item child focused experience questions has provided insight to the impacts of food insecurity for children. The USDA-HFSSM tool identifies the primary determinant of food insecurity, being limited financial resources. There are also some limitations to this study. This study was cross-sectional in design and the analyses are descriptive in nature and self reported. In addition to questions pertaining to food security this study was part of a wider study exploring the impacts of COVID-19 on women’s diet, mental health, and physical activity. Further, pre-COVID was not defined in the survey, this may have impacted on the reporting of pre-COVID variables. Due to the number of survey questions and the time to complete the survey, there was limited scope to explore reasons beyond the financial causes of food insecurity, such as food cost, food availability and physical access to food. These factors are particularly pertinent in regional and remote areas of Australia [49];,and also may have been important factors in the context of lockdown restrictions limiting the travel distances from home and impacting access to healthy, affordable and walkable grocery and food outlets [50]. Whilst measures were undertaken to ensure sampling targeted underrepresented groups to align with population characteristics, an online survey may exclude those with low literacy levels and poor internet access. Further, ethnicity did not specifically identify indigenous peoples or those from a Culturally and Linguistically Diverse background. This study used a well-established database, which may impact the diversity of food security experience reported and therefore the generalisability of these results.

## 5. Conclusions

Using a comprehensive and validated measure of food security, we demonstrated an increased prevalence of food insecurity during the COVID-19 pandemic in Australian women. This study identified that 20% of women were living in households experiencing food insecurity, which is considerably higher than pre-COVID-19 prevalence. Further, there were 13.7% households who were marginally food secure, who experienced high levels of worry and concern about running out of food. Food insecurity was experienced at all levels of pre-COVID-19 income levels. Mental health, housing status, income level and change of employment status due to COVID-19 were shown to be associated with food security status. This study has also highlighted the child experience of food insecurity within households during the pandemic. With ongoing restrictions and lockdowns, consideration needs to be given to the financial support provided to people and regions at risk of food insecurity. Food insecurity, poor mental health, inadequate income and lack of adequate housing existed before COVID-19. However, this study highlights the additional impact of COVID-19 restrictions on each of these factors, which has increased the prevalence of food insecurity and highlighted this as an issue previously hidden/underreported. The risk of COVID-19 lockdowns and the subsequent economic implications are not easing any time soon; if left ignored, food security status may continue to deteriorate. This study supports the need for a range of policy and programmatic responses that address mental health, financial, employment and housing support to mitigate the impacts of ongoing COVID-19 restrictions in Australia, particularly for women and children. As the pandemic continues, this study also provides a view of what to expect in the future regarding household food security status in regard to COVID-19.

## Figures and Tables

**Figure 1 nutrients-13-04262-f001:**
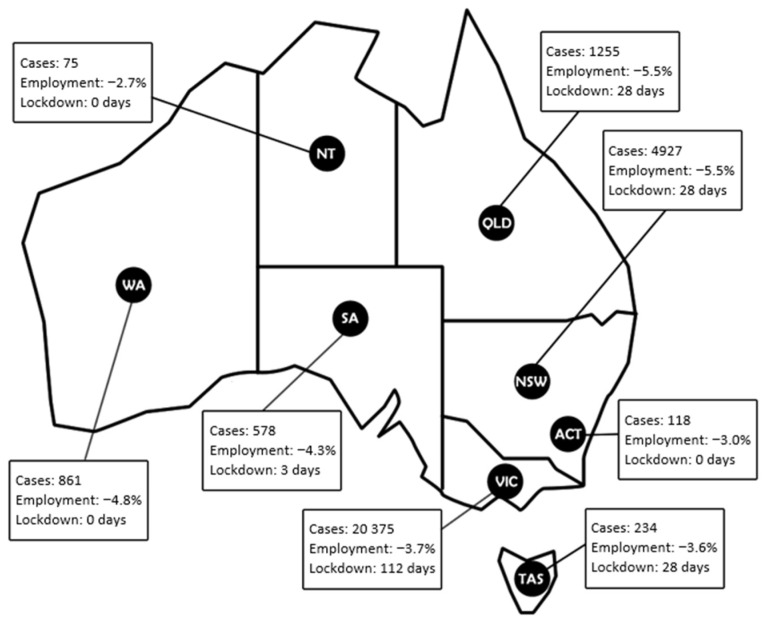
COVID-19 cases during 2020 and seasonally adjusted change in percentage employment in May 2020 by state or territory. Abbreviations: ACT: Australian Capital Territory; NSW: New South Wales; NT: Northern Territory; QLD: Queensland; SA: South Australia; TAS: Tasmania; VIC: Victoria; WA: Western Australia [17].

**Table 1 nutrients-13-04262-t001:** Food security status by socio-demographic characteristics during COVID-19 in Australia in 2020.

Demographic	Category	Food Security Status *n* (%) *n* = 1005				*p*-Value *
		High (*n* = 670)	Marginal (*n* = 138)	Low (*n* = 119)	Very Low (*n* = 78)	
**State/Territory**	Australian Capital Territory	20 (95.2)	0 (0.0)	0 (0.0)	1 (4.8)	0.058
	New South Wales	202 (65.0)	44 (14.1)	44 (14.1)	21 (6.8)	
	Northern Territory	2 (66.7)	0 (0.0)	0 (0.0)	1 (33.3)	
	Queensland	128 (64.6)	36 (18.2)	22 (11.1)	12 (6.1)	
	South Australia	47 (69.1)	5 (7.4)	12 (17.6)	4 (5.9)	
	Tasmania	21 (72.4)	3 (10.3)	5 (17.2)	0 (0)	
	Victoria	180 (64.7)	40 (14.4)	29 (10.4)	29 (10.4)	
	Western Australia	70 (72.2)	10 (10.3)	7 (7.2)	10 (10.3)	
** Regional/Remote **	Inner Regional Australia	67 (64.4)	17 (16.3)	10 (9.6)	10 (9.6)	0.705
	Major Cities of Australia	563 (66.5)	118 (13.9)	102 (12.0)	64 (7.6)	
	Outer Regional Australia	30 (71.4)	3 (7.1)	5 (11.9)	4 (9.5)	
	Remote Australia	9 (81.8)	0 (0)	2 (18.2)	0 (0)	
**Age**	18–24	108(68.4)	21 (13.3)	14 (8.9)	15(9.5)	0.215
	25–34	228 (65.3)	44 (12.6)	45 (12.9)	32 (9.2)	
	35–44	219 (66.6)	49 (14.9)	46 (14.0)	15 (4.6)	
	45–50	115 (68.0)	24 (14.2)	14 (8.3)	16 (9.5)	
** Current Marital Status **	Single	254 (61.2)	58 (14.0)	57 (13.7)	46 (11.1)	0.005
	Married or de facto	402 (70.0)	80 (13.9)	61 (10.6)	31 (5.4)	
	I prefer not to say	14 (87.5)	0 (0)	1 (6.3)	1(6.3)	
** Number of Children **	0 children	419 (73.1)	65 (11.3)	46 (8.0)	43 (7.5)	<0.01
	1 child	89 (56.3)	26 (16.5)	26 (16.5)	17 (10.8)	
	2 children	116 (61.1)	27(14.2)	31 (16.3)	16 (8.4)	
	3 or more children	46 (55.4)	19 (22.9)	16 (19.3)	2 (2.4)	
**Highest Level of Completed Education**	Primary/elementary school or less	2 (40.0)	0 (0)	1 (20.0)	2 (40.0)	0.003
	Secondary/high school	112 (58.0)	27 (14.0)	29 (15.0)	25 (13.0)	
	TAFE	156 (65.0)	37 (15.4)	32 (13.3)	15 (6.3)	
	University/Post-graduate degree	397 (70.8)	71 (12.7)	57 (10.2)	36 (6.4)	
** Self-identified cultural/ethnic group **	Oceanian (Australian peoples, New Zealand peoples, Pacific Islanders)	367 (65.4)	75 (13.4)	68 (12.1)	51 (9.1)	0.830
	North-west European (British, Irish, Western European, Northern European)	106 (66.7)	25 (15.7)	17 (10.7)	11 (6.9)	
	South East Asian	56 (70.0)	11 (13.8)	11 (13.8)	2 (2.5)	
	Southern and Eastern European (Southern European, South Eastern European, Eastern European)	47 (65.3)	13 (18.1)	9 (12.5)	3 (4.2)	
	North East Asian	21 (60.0)	5 (14.3)	5 (14.3)	4 (11.4)	
	North African and Middle Eastern (Arab, Jewish, Peoples of the Sudan, other North African and Middle Eastern)	13 (72.2)	1 (5.6)	3 (16.7)	1 (5.6)	
	North American	10 (90.9)	1 (9.1)	0 (0)	0 (0)	
	Southern and Central Asian	9 (81.8)	1 (9.1)	1 (9.1)	0 (0)	
	Southern and East African	6 (100.0)	0 (0)	0 (0)	0 (0)	
	South American	3 (42.9)	1 (14.3)	2 (28.6)	1 (14.3)	
	I prefer not to say	32(71.1)	5 (11.1)	3 (6.7)	5 (11.1)	
** Housing status **	Own home	350 (71.1)	62 (12.6)	54 (11.0)	26 (5.3)	0.006
	Rented home	178 (58.6)	46 (15.1)	41 (13.5)	39 (12.8)	
	Living with family	136 (68.7)	27 (13.6)	22 (11.1)	13 (6.6)	
	Emergency accommodation (hostel, B&B, hotel)	0 (0)	1 (50.0)	1 (50.0)	0 (0)	
	I prefer not to say	6 (66.7)	2 (22.2)	1 (11.1)	0 (0)	
**Mental Health Score (K10)**	K10 ≥ 20	257 (52.4)	88 (18.0)	81 (16.5)	64 (13.1)	<0.001
** Pre COVID-19 employment **	Full-time employment	372 (72.8)	57 (11.2)	47 (9.2)	35 (6.8)	0.014
	Casual employment	47 (65.3)	9 (12.5)	8 (11.1)	8 (11.1)	
	Government assistance	18 (62.1)	4 (13.8)	5 (17.2)	2 (6.9)	
	Government disability support	9 (50.0)	4 (22.2)	2 (11.1)	3 (16.7)	
	Homemaker	57 (56.4)	18 (17.8)	13 (12.9)	13 (12.9)	
	Part-time employment	115 (62.2)	34 (18.4)	29 (15.7)	7 (3.8)	
	Retired	2 (50.0)	0 (0.0)	1 (25.0)	1 (25.0)	
	Student	42 (62.7)	7 (10.4)	12 (17.9)	6 (9.0)	
	Pref not say	8 (44.4)	5 (27.8)	2 (11.1)	3 (16.7)	
** Pre COVID-19 household income (AUD) **	0–24,999	19 (32.2)	15 (25.4)	12 (20.3)	13 (22.0)	<0.001
	25,000–49,999	42 (46.2)	15 (16.5)	21 (23.1)	13 (14.3)	
	50,000–74,999	70 (54.7)	17 (13.3)	23 (18.0)	18 (14.1)	
	75,00–99,999	103 (65.2)	26 (16.5)	14 (8.9)	15 (9.5)	
	100,000–124,999	86 (69.4)	18 (14.5)	14 (11.3)	6 (4.8)	
	125,000–149,999	72 (76.6)	11 (11.7)	7 (7.4)	4 (4.3)	
	>150,000	178 (84.4)	19 (9.0)	9 (4.3)	5 (2.4)	
** Change in employment status due to COVID-19 **	Yes	127 (50.4)	39 (15.5)	43 (17.1)	43 (17.1)	<0.001
	No	543 (72.1)	99 (13.1)	76 (10.1)	35 (4.6)	
** Employment status due to COVID-19 **	Full-time employment	27(54.0)	3 (6.0)	7 (14.0)	13 (26.0)	0.602
	Casual employment	19 (46.3)	11 (26.8)	6 (14.6)	5 (12.2)	
	Government assistance e.g., family payments	1 (11.10)	3 (33.3)	1 (11.1)	4 (44.4)	
	Government assistance e.g., Job Keeper	10 (45.5)	4 (18.2)	6 (27.3)	2 (9.1)	
	Government assistance e.g., Job Seeker	17 (53.1)	5 (15.6)	5 (15.6)	5 (15.6)	
	Government disability support	2 (50.0)	1 (25.0)	0 (0.0)	1 (25.0)	
	Homemaker	5 (50.0)	2 (20.0)	2 (20.0)	1 (10.0)	
	Part-time employment	30 (53.6)	8 (14.3)	11 (19.6)	7 (12.5)	
	Retired	1 (100.0)	0 (0.0)	0 (0.0)	0 (0)	
	Student	9 (50.0)	1 (5.6)	4 (22.2)	4 (22.2)	
	I don’t know/I prefer not to answer	6 (66.7)	1 (11.1)	1 (11.1)	1 (11.1)	

Table 1 footnotes: * *p*-value calculated via chi-squared.

**Table 2 nutrients-13-04262-t002:** Responses to the 18-item USDA Household Food Security Survey across food security status during COVID-19 in Australia in 2020.

USDA Household Food Security Survey Module Question	Response Option	Food Security Status *n* (%)				*p* Value *
		High	Marginal	Low	Very Low	
**Adult Question Items in the Last 6 Months**						
(I/We) worried whether (my/our) food would run out before (I/we) got money to buy more	Sometimes true/often true	0 (0.0)	73 (9.9)	10 (1.4)	4 (0.5)	<0.001
	Never true	652 (88.2)	19 (11.3)	78 (46.4)	71 (42.30)	
	I don’t know/I prefer not to answer	18 (78.3)	2 (8.7)	3 (13.0)	0 (0.0)	
The food that (I/we) bought just didn’t last and (I/we) didn’t have money to get more	Sometimes true/often true	0 (0.0)	19 (11.3)	78 (46.4)	71 (42.3)	<0.001
	Never true	655 (80.3)	116 (14.2)	39 (4.8)	6 (0.7)	
	I don’t know/I prefer not to answer	15 (71.4)	3 (14.3)	2 (9.5)	1 (4.8)	
(I/we) couldn’t afford to eat balanced meals	Sometimes true/often true	0 (0.0)	60 (25.9)	98 (42.2)	74 (31.9)	<0.001
	Never true	653 (86.7)	76 (10.1)	21 (2.8)	3 (0.4)	
	I don’t know/I prefer not to answer	17 (85.0)	2 (10.0)	0 (0.0)	1 (5.0)	
In the last 6 months since last did you or other adults in your household ever cut the size of your meals or skip meals because there wasn’t enough money for food	Yes	N/A	1 (0.9)	34 (31.5)	73 (67.6)	<0.001
	No	N/A	111 (57.5)	78 (40.4)	4 (2.1)	
	I don’t know/I prefer not to answer	N/A	4 (54.5)	6 (9.1)	1 (36.4)	
How often did this happen—almost every month some months but not every month or in only 1 or 2 months	Only 1 or 2 months	N/A	1 (3.7)	17 (63.0)	9 (33.3)	0.002
	Almost every month/Some months but not every month	N/A	0 (0.0)	16 (20.3)	63 (79.7)	
	I don’t know/I prefer not to answer	N/A	0 (0.0)	1 (50.0)	1 (50.0)	
In the last 6 months did you ever eat less than you felt you should because there wasn’t enough money for food?	Yes	N/A	7 (5.7)	44 (36.1)	71 (58.2)	<0.001
	No	N/A	106 (58.6)	69 (38.1)	6 (3.3)	
	I don’t know/I prefer not to answer	N/A	3 (33.3)	5 (55.6)	1 (11.1)	
In the last 6 months were you every hungry but didn’t eat because there wasn’t enough money for food?	Yes	N/A	1 (1.1)	27(29.0)	65 (69.9)	<0.001
	No	N/A	114 (54.00)	86 (40.8)	11 (5.2)	
	I don’t know/I prefer not to answer	N/A	1 (12.5)	5 (62.5)	2 (25.0)	
In the last 6 months did you lose weight because there wasn’t enough money for food?	Yes	N/A	1 (1.7)	10 (16.9)	48 (81.4)	<0.001
	No	N/A	115 (48.9)	99 (42.1)	21 (8.9)	
	I don’t know/I prefer not to answer	N/A	0 (0.0)	9 (50.0)	9 (50.0)	
In the last 6 months did (you/you or other adults in your household) ever not eat for a whole day because there wasn’t enough money for food?	Yes	N/A	0 (0.0)	2 (4.7)	41 (95.3)	<0.001
	No	N/A	9 (8.5)	60 (56.6)	37 (34.9)	
	I don’t know/I prefer not to answer	N/A	1 (25.0)	3 (75.0)	0 (0.0)	
How often did this happen—almost every month some months but not every month or in only 1 or 2 months?	Only 1 or 2 months	N/A	N/A	2 (22.2)	7 (77.8)	0.004
	Almost every month/Some months but not every month	N/A	N/A	0 (0.00)	34 (100)	
**Child Question Items in the last 6 months**						
I relied on only a few kinds of low-cost food to feed my child/the children because we were running out of money to buy food.	Often true/Sometimes true	0 (0.0)	35 (28.0)	56 (44.8)	34 (27.2)	<0.001
	Never true	245 (82.5)	36 (12.1)	15 (5.1)	1 (0.3)	
	I don’t know/I prefer not to answer	6 (66.7)	1 (11.1)	2 (22.2)	0 (0.0)	
I couldn’t feed my child/the children a balanced meal because we couldn’t afford that.	Often true/Sometimes true	0 (0.0)	2 (7.4)	13 (48.1)	12 (44.0)	<0.001
	Never true	87 (68.5)	24 (18.9)	12 (9.4)	4 (3.1)	
	I don’t know/I prefer not to answer	2 (50.0)	0 (0.0)	1 (25.0)	1 (25.0)	
My child was/The children were not eating enough because we just couldn’t afford enough food.	Often true/Sometimes true	0 (0.00)	0 (0.0)	8 (47.1)	9 (52.9)	<0.001
	Never true	87 (64.00)	26 (19.1)	16 (11.8)	7 (5.1)	
In the last 6 months did you ever cut the size of your childs/any of the children’s meals because there wasn’t enough money for food?	Yes	N/A	3 (15.8)	2 (10.5)	14 (73.7)	<0.001
	No	N/A	34 (31.5)	55 (50.9)	19 (17.6)	
In the last 6 months did any of the children ever skip meals because there wasn’t enough money for food?	Yes	N/A	0 (0.0)	0 (0.0)	12 (100.0)	0.001
	No	N/A	37 (31.9)	57 (49.1)	22 (19.0)	
How often did this happen—almost every month some months but not every month or in only 1 or 2 months?	Only 1 or 2 months	N/A	N/A	N/A	2 (100.0)	<0.001
	Almost every month/Some months but not every month	N/A	N/A	N/A	10 (100.0)	
In the last 6 months was your child/were the children ever hungry but you just couldn’t afford more food?	Yes	0 (0.00)	0 (0.0)	11 (100)	0 (0.0)	<0.001
	No	37 (31.9)	56 (48.3)	23 (19.8)	37 (31.9)	
	I don’t know/I prefer not to answer	0 (0.0)	2 (100)	0 (0.00)	0 (0.0)	
In the last 6 months did your child/any of the children ever not eat for a whole day because there wasn’t enough money for food?	Yes	0	0	9 (100)	0	<0.001
	No	37 (31.9)	56 (48.3)	23 (19.8)	37 (31.9)	
	I don’t know/I prefer not to answer	0 (0.0)	2 (50.0)	2 (50.0)	0 (0.0)	

Table 2 footnotes: * *p*-value calculated via chi-squared. N/A represents questions that are not presented in the tool due to skip logic.

**Table 3 nutrients-13-04262-t003:** Association between risk factors and food insecurity status during COVID-19 in Australia in 2020 multivariable logistic regression.

	Marginal Food Security		Low Food Security		Very Low Food Security	
	OR (95%CI)	*p*-Value	OR (95%CI)	*p*-Value	OR (95%CI)	*p*-Value
**State**						
State (all other states)	Ref	Ref	Ref	Ref	Ref	Ref
NSW	1.158 (0.643–1.934)	0.576	1.144 (0.643–2.037)	0.647	0.718 (0.339–1.522)	0.388
VIC	1.090 (0.646–1.839)	0.746	0.714 (0.384–2.135)	0.285	1.171 (0.593–2.314)	0.649
**Current marital status**						
Marital status (married/de facto)	Ref	Ref	Ref	Ref	Ref	Ref
Single	0.910 (0.526–1.572)	0.734	1.163 (0.628–2.153)	0.631	1.516 (0.751–3.061)	0.245
I prefer not to say	N/A	N/A	N/A	N/A	4.688 (0.299–73.591)	0.271
**Number of children**						
No children	Ref	Ref	Ref	Ref	Ref	Ref
1 child	2.241 (1.238–4.055)	0.008	3.938 (1.997–7.765)	<0.001	2.578 (1.172–5.675)	0.019
2 children	1.932 (1.034–3.608)	0.039	3.811 (1.921–7.561)	<0.001	2.434 (1.045–5.673)	0.039
>3 children	3.587 (1.677–7.672)	<0.001	6.243 (2.593–15.032)	<0.001	0.305 (0.034–2.715)	0.287
**Education**						
Education (university/post-grad)	Ref	Ref	Ref	Ref	Ref	Ref
TAFE	0.870 (0.509–1.486)	0.609	0.911 (0.495–1.676)	0.765	0.631 (0.294–1.352)	0.236
High School	1.116 (0.623–2.003)	0.712	1.188 (0.621–2.274)	0.603	1.312 (0.621–2.774)	0.477
Primary school	N/A	N/A	1.308 (0.056–30.476)	0.867	2.002 (0.117–34.327)	0.632
**Housing tenure**						
Own Home	Ref	Ref	Ref	Ref	Ref	Ref
Rented home	1.238 (0.750–20.41)	0.404	1.015 (0.578–1.782)	0.959	2.102 (1.091–4.048)	0.026
Living with family	1.505 (0.805–2.811)	0.200	0.631 (0.290–1.374)	0.246	0.711 (0.275–1.842)	0.483
Emergency accommodation	N/A	N/A	N/A	N/A	N/A	N/A
**Mental health**						
Mental health above 20	2.51 (1.612 -3.909)	<0.001	5.360 (3.097–9.278)	<0.001	7.065 (3.404–14.662)	<0.001
**Pre-COVID19 employment**						
Job (full time)	Ref	Ref	Ref	Ref	Ref	Ref
Casual	0.508 (0.183–1.410)	0.194	0.803 (0.278–0.316)	0.684	0.939 (3.18–2.773)	0.910
Government assistance	0.365 (0.091 (1.469)	0.156	0.789 (0.212–2.941)	0.724	0.383 (0.070–2.095)	0.268
Government disability support	1.460 (0.296–7.207)	0.643	1.050 (0.159–6.920)	0.959	1.740 (0.266–11.383)	0.563
Homemaker	0.825 (0.363–1.879)	0.647	0.619 (0.226–1.691)	0.349	2.212 (0.735 -6.656)	0.158
Part-time	1.134 (0.641–2.008)	0.666	1.293 (0.670–2.495)	0.444	0.484 (0.184–1.271)	0.141
Retired	N/A	N/A	N/A	N/A	1.927 (0.113–32.857)	0.650
Student	0.636 (0.217–1.859)	0.408	1.573 (0.553 -4.472)	0.395	1.178 (0.346–4.013)	0.793
I don’t know/I prefer not to say	0.833 (0.178–4.383)	0.879	1.344 (0.219–8.257)	0.750	0.435 (0.042–4517)	0.485
**Pre-COVID19 income**						
Income > $150,000	Ref	Ref	Ref	Ref	Ref	Ref
$125,000–149,999	1.510 (0.662–3.444)	0.327	2.040 (0.645–6.449)	0.225	1.312 (0.302–5.693)	0.717
$100,000–$124,999	2.064 (1.000–4.265)	0.050	3.539 (1.355–9.242)	0.010	1.867 (0.506–6.887)	0.348
$75,00–$99,999	2.187 (1.095–4.367)	0.027	2.729 (10.35–7.195)	0.042	3.036 (0.971–9.661)	0.056
$50,000–$74,999	2.198 (0.996–4.848)	0.051	7.424 (2.871–19.195)	<0.001	4.407 (1.38–13.992)	0.012
$25,000–$49,999	4.371 (1.748–10.932)	0.002	10.877 (3.692–32.042)	<0.001	6.166 (1.689–22.512)	0.006
$0–$24,999	8.896 (3.127–25.308)	<0.001	10.293 (2.836–37.359)	<0.001	8.976 (2.175–37.033)	0.002
**Change in employment status**						
Has your employment status changed since COVID-19—Yes	1.611 (0.985–2.636)	0.57	2.222 (1.306–3.779)	0.003	6.513 (3.475–12.207)	<0.001

Table 3 footnotes: All ORs are in reference to high food security. Covariates included in the model: State, marital status, children, education, pre-COVID-19 employment, pre COVID-19 income and change to employment status due to COVID-19. Abbreviations: Ref = reference, OR = odds ratio and 95%CI = 95% confidence interval. N/A represents results that were unable to be displayed due to low respondent numbers.

## Data Availability

The data presented in this study are available on request from the corresponding author. The data are not publicly available due to ethical approval.

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
