# Peer review of "Food Insecurity Prevalence, Severity and Determinants in Australian Households during the COVID-19 Pandemic from the Perspective of Women"

_nutrients, 2021, doi:10.3390/nu13124262_

Round 1
Reviewer 1 Report
This is an interesting research. I have included my detailed feedback and suggestions for revision in the attached document.
Thank you for conducting this interesting research in an important area, where research are limited. I have provided some comment and suggestions about how the manuscript can be improved.
Analysis and presentation of results:
- It is okay to present the descriptive results with the various category of responses, even if the cell frequency is quite low. However, for table 2 regression analysis some of these categories (e.g. retired, n =1; pre Covid income categories; >2 children) can be combined together or excluded from analysis. Similar logic apply to some of the ‘prefer not to say’ responses, which have very low number of responses. Collapsing categories, or excluding groups which do not provide much additional information (e.g. prefer not to say) would make the results more useful.
- Given that the research is more exploratory in nature and did not include a representative sample, the logistic regression for all the categories of food security (e.g. very low food security, total n =78) is not so meaningful. Within a food security group (e.g., low or very low food security) , breakdown of the various categorical variable included in the regression would mean the numbers compared are very small and there will be issues of lack of statistical power for a meaningful comparison.
- I suggest that the authors combine very low and low food security as food insecurity and combine marginal and high food security as food security. The authors will be able to find references for this categorisation. They can then have the outcome as food insecurity (food secure =0, food insecure =1) to perform a logistic regression which will reflect which factors are associated with food insecurity. This way the analysis will retain better statistical power.
Results:
- The description of results on line 179-181 seems inappropriate. It should basically say that food security status varied significantly by pre-Covid employment status. Higher percentage of those on government disability assistance or were homemaker had low or very low food security.
- Change in employment status results description (line 182-183): To give a complete picture the authors need to present results for those who did not have change in employment. The chi-square test is assessing whether there is a significant difference between food security levels for those who had change in employment versus those who did not have a change. In other words, it is assessing whether there is a significant association between change in employment type and food security levels. The explanation of the results in terms of what this association meant seems incorrect. The percentage would add up to 100% by food security status, for both those who had change in employment status and those who did not have change in employment status. The association means that there was higher percentage of women with low food security among those who had a change in employment status compared with those who did not have change. It is not that food insecurity increased, rather percentage of people with low or very low food security was higher among those who had change in employment.
- Table 3 results (% shown by level of food security) are difficult to understand (which way they add to 100%). Ideally, I think since the authors are using a validated measure of food security, it is natural that questions related to severe insecurity experiences will be more frequent among the low / very low food security category. Therefore, there is no reason why a test of association should be done for the 18 item food security questions and level of food security. These questions are coming as a set of validated questions to assess food security and thus an association is expected. The authors should use this table 3 only as a descriptive result to describe the context and present column percentage, rather than row percentage. Here, authors can also exclude prefer not say responses from analysis. This table should also come before the regression results and just presented as a frequency table within each category of food security levels. For these results, no P value are required or would be very useful.
Author Response
Analysis and presentation of results:
- It is okay to present the descriptive results with the various category of responses, even if the cell frequency is quite low. However, for table 2 regression analysis some of these categories (e.g. retired, n =1; pre Covid income categories; >2 children) can be combined together or excluded from analysis. Similar logic apply to some of the ‘prefer not to say’ responses, which have very low number of responses. Collapsing categories, or excluding groups which do not provide much additional information (e.g. prefer not to say) would make the results more useful.
Author Response: Thank you for this comment for our consideration. We appreciate your point about low n categories used within the regression. However, we believe it is important to highlight the nuances in the data collected, showing the breadth of the experience of food insecurity across different demographics.
- Given that the research is more exploratory in nature and did not include a representative sample, the logistic regression for all the categories of food security (e.g. very low food security, total n =78) is not so meaningful. Within a food security group (e.g., low or very low food security) , breakdown of the various categorical variable included in the regression would mean the numbers compared are very small and there will be issues of lack of statistical power for a meaningful comparison. I suggest that the authors combine very low and low food security as food insecurity and combine marginal and high food security as food security. The authors will be able to find references for this categorisation. They can then have the outcome as food insecurity (food secure =0, food insecure =1) to perform a logistic regression which will reflect which factors are associated with food insecurity. This way the analysis will retain better statistical power.
Author Response: Thank you for this comment. We appreciate the feedback. However, the reason this has analysis was conducted to highlight the spectrum of the experience of food security. This is the value of using such a tool as the USDA 18item. There is limited reporting of the severity categories. Therefore, we have included an additional component to this manuscript that provides the readers with a more nuanced understanding of the food security experience.
Results:
- The description of results on line 179-181 seems inappropriate. It should basically say that food security status varied significantly by pre-Covid employment status. Higher percentage of those on government disability assistance or were homemaker had low or very low food security.
Author Response: Thank you for this comment. We have amended this text inline with your suggestion which is now available on lines 229-231:
‘As food insecurity severity increased, so did the proportion of women were employed casually, be on government disability assistance payments or a homemaker (p=0.014) prior to the COVID-19 pandemic’
- Change in employment status results description (line 182-183): To give a complete picture the authors need to present results for those who did not have change in employment. The chi-square test is assessing whether there is a significant difference between food security levels for those who had change in employment versus those who did not have a change. In other words, it is assessing whether there is a significant association between change in employment type and food security levels. The explanation of the results in terms of what this association meant seems incorrect. The percentage would add up to 100% by food security status, for both those who had change in employment status and those who did not have change in employment status. The association means that there was higher percentage of women with low food security among those who had a change in employment status compared with those who did not have change. It is not that food insecurity increased, rather percentage of people with low or very low food security was higher among those who had change in employment.
Author Response: Thank you for this comment. This has now been addressed in Table 1 as per your recommendation. Further, please find this amended text on lines 232-234: ‘There was an association between food security and changes in employment status due to COVID-19; whereby, those that reported changes in their employment due to COVID-19 were more likely to experience food insecurity (p<0.001).’
- Table 3 results (% shown by level of food security) are difficult to understand (which way they add to 100%). Ideally, I think since the authors are using a validated measure of food security, it is natural that questions related to severe insecurity experiences will be more frequent among the low / very low food security category. Therefore, there is no reason why a test of association should be done for the 18 item food security questions and level of food security. These questions are coming as a set of validated questions to assess food security and thus an association is expected. The authors should use this table 3 only as a descriptive result to describe the context and present column percentage, rather than row percentage. Here, authors can also exclude prefer not say responses from analysis. This table should also come before the regression results and just presented as a frequency table within each category of food security levels. For these results, no P value are required or would be very useful.
Author Response: Thank you for this comment. We can appreciate your point about the validity of including p-values and the response variables in Table 2 (previously 3). However, the inclusion of response variables is in line with previously published papers in the area (Kent et al). In addition to what Kent et al have published, our paper includes the 18-item tool, which includes the adult and child experience in the household. This has not been previously reported in an Australian setting and hence important to highlight.
Reviewer 2 Report
The present cross-sectional study aimed to investigate the prevalence, severity and socio-demographic predictors of household food insecurity from the perspective of Australian women of reproductive age (18- 50 years) during the COVID-19 pandemic during 2020. This is an interesting report and of potential interest to the readers in the field. I have the following minor concerns.
- Explain why you choose to study the female population in your investigation.
- A total of n=1005 women were recruited. Please add a flowchart to demonstrate the way you enrolled the participants in your study. Did you exclude any participants and please mention the compliance of all participants. Did they all completed the whole survey?
- All participants enrolled in the study completed questionnaires for data collection. The results from asking the participants may not be so reliable. Keep asking the participants a series of questionnaires may get a random distribution of percentage. Please discuss.
- Are there any happened or possible complications resulting from participants joining the study?
Author Response
The present cross-sectional study aimed to investigate the prevalence, severity and socio-demographic predictors of household food insecurity from the perspective of Australian women of reproductive age (18- 50 years) during the COVID-19 pandemic during 2020. This is an interesting report and of potential interest to the readers in the field. I have the following minor concerns.
- Explain why you choose to study the female population in your investigation.
- A total of n=1005 women were recruited. Please add a flowchart to demonstrate the way you enrolled the participants in your study. Did you exclude any participants and please mention the compliance of all participants? Did they all completed the whole survey?
Author Response: Thank you for this comment. As this is a cross sectional study it is difficult to see how a flow chart would be helpful. However, we have included the following information about the exclusion of participants that did not fill in the survey adequately. This information can be found on lines 146-148 of the manuscript:
‘Data screening and cleaning ensured data usability and an integrity script allowed discarding of surveys with less than 10% completion (n=508).’
Further, we have amended the first line of the results to more clearly articulate this (lines 161-162)
‘A total of n=1005 women were included in the analysis’
- All participants enrolled in the study completed questionnaires for data collection. The results from asking the participants may not be so reliable. Keep asking the participants a series of questionnaires may get a random distribution of percentage. Please discuss.
Author Response: We agree that a cross-sectional and self-reported nature is a limitation of this study. We have included this in the limitations of the study in lines 418-419
‘This study was cross-sectional in design and the analyses are descriptive in nature’
- Are there any happened or possible complications resulting from participants joining the study?
Author Response: This was a cross-sectional, non-interventional study that collected anonymous data. Therefore, there was no complications of the study.
Reviewer 3 Report
I read the manuscript entitled “Food insecurity prevalence, severity and determinants in Aus-2 tralian households during the COVID-19 Pandemic from the 3 perspective of women” by Kleve et al with great interest, and even though there is a previous study with similar scheme (reference #24) from Australia (actually Tasmania), I found that as a women perspective and using nationwide sampling method, this current manuscript has meaningful importance and may attract some citation and visibility after publication. I found the manuscript well written and have only a few minor-to-moderate suggestions here.
- Abstract, introduction, and methods are well written.
- Results section consists with 3 main parts 1) demographic factors, 2) factors associated with food insecurity, 3) questionnaire items and food security status. I suggest authors may use subheadings for results, for example, “Basic demographic characteristics”, etc. Also, as the third part on item response and food security status is supporting evidence for internal validity of the questionnaire, I suggest change the order of presentation of 3) and 2) in results section, for logical process.
- Data shown that geographic location of participants did not significantly affect the state of food insecurity, and authors suspected that adverse effect of prolonged lockdown for food security might be attenuated by the Federal support in discussion. While this might be true in 2020, because of really prolonged and draconian lockdowns in Australia, compared to other developed countries, I wonder whether Federal financial support will be sustainable and enough to compensate disproportionately affected population who are already insecure but further socioeconomically adversely affected with lockdowns (especially people in travel, hospitality sector, and employed with flexible contracts). I suggest authors to add up some remarks on existing uncertainty afterwards the time point the questionnaire was administered in 2020.
- Line 171, 42.7% (n=42) reported a house-171 hold income greater than AUD$100,000 per year – For 42.7%, it should be >420, I think it would be a typo. In Table 3, ‘AD1a’ seems to be a typo. ‘6.Patents –this section----manuscript’ can be deleted.
Author Response
Reviewer 1:
I read the manuscript entitled “Food insecurity prevalence, severity and determinants in Aus-2 tralian households during the COVID-19 Pandemic from the 3 perspective of women” by Kleve et al with great interest, and even though there is a previous study with similar scheme (reference #24) from Australia (actually Tasmania), I found that as a women perspective and using nationwide sampling method, this current manuscript has meaningful importance and may attract some citation and visibility after publication. I found the manuscript well written and have only a few minor-to-moderate suggestions here. Abstract, introduction, and methods are well written.
- Results section consists with 3 main parts 1) demographic factors, 2) factors associated with food insecurity, 3) questionnaire items and food security status. I suggest authors may use subheadings for results, for example, “Basic demographic characteristics”, etc. Also, as the third part on item response and food security status is supporting evidence for internal validity of the questionnaire, I suggest change the order of presentation of 3) and 2) in results section, for logical process.
Author Response: Thank you for this suggestion. We have reorganised the results inline with this suggestion.
- Data shown that geographic location of participants did not significantly affect the state of food insecurity, and authors suspected that adverse effect of prolonged lockdown for food security might be attenuated by the Federal support in discussion. While this might be true in 2020, because of really prolonged and draconian lockdowns in Australia, compared to other developed countries, I wonder whether Federal financial support will be sustainable and enough to compensate disproportionately affected population who are already insecure but further socioeconomically adversely affected with lockdowns (especially people in travel, hospitality sector, and employed with flexible contracts). I suggest authors to add up some remarks on existing uncertainty afterwards the time point the questionnaire was administered in 2020.
Author Response: Thank you for this reflection. We agree this is an important consideration. The manuscript includes a remark about this on lines 403-406
‘This may change as Australia continues to experience strict restrictions which have impacted industries such as retail, beauty, recreation and hospitality without financial support from the Federal or State Governments’
- Line 171, 42.7% (n=42) reported a house-171 hold income greater than AUD$100,000 per year – For 42.7%, it should be >420, I think it would be a typo.
Author Response: Thank you – this has now been amended.
- In Table 3, ‘AD1a’ seems to be a typo.
Author Response: Thank you – this has now been amended.
- ‘6.Patents –this section----manuscript’ can be deleted.
Author Response: Thank you – this has now been amended.